# Ward-level factors associated with methicillin-resistant *Staphylococcus aureus* acquisition–an electronic medical records study in Singapore

Zaw Myo Tun[1]*, Dale A. Fisher[2,3], Sharon Salmon[4], Clarence C. Tam[1,5]

**1** Saw Swee Hock School of Public Health, National University of Singapore, Singapore, Singapore, **2** Division of Infectious Diseases, National University Hospital, Singapore, Singapore, **3** Yong Loo Lin School of Medicine, National University of Singapore, Singapore, Singapore, **4** UNSW Medicine, University of New South Wales, Sydney, Australia, **5** London School of Hygiene and Tropical Medicine, London, United Kingdom

* zawmyotun@nus.edu.sg

## Abstract

### Background

Methicillin-Resistant *Staphylococcus aureus* (MRSA) is endemic in hospitals worldwide. Intrahospital transfers may impact MRSA acquisition risk experienced by patients. In this study, we investigated ward characteristics and connectivity that are associated with MRSA acquisition.

### Methods

We analysed electronic medical records on patient transfers and MRSA screening of in-patients at an acute-care tertiary hospital in Singapore to investigate whether ward characteristics and connectivity within a network of in-patient wards were associated with MRSA acquisition rates over a period of four years.

### Results

Most patient transfers concentrated in a stable core network of wards. Factors associated with increased rate of MRSA acquisition were MRSA prevalence among patients transferred from other wards (rate ratio (RR): 7.74 [95% confidence interval (CI): 3.88, 15.44], additional 5 percentage point), critical care ward (RR: 1.72 [95% CI: 1.09, 2.70]) and presence of MRSA cohorting beds (RR: 1.39 [95% CI: 1.03, 1.90]). Oncology ward (RR: 0.66 [95% CI: 0.46, 0.94]) (compared to medical ward), and median length of stay (RR: 0.70 [95% CI: 0.55, 0.90], additional 1.5 days) were associated with lower acquisition rates. In addition, we found evidence of interaction between MRSA prevalence among patients transferred from other wards and weighted in-degree although the latter was not associated with MRSA acquisition after controlling for confounders.

**Data Availability Statement:** All relevant data are within the paper and its Supporting Information files.

**Funding:** The authors received no specific funding for this work.

**Competing interests:** The authors have declared that no competing interests exist.

## Conclusion

Wards with higher MRSA prevalence among patients transferred from other wards were more likely to have higher MRSA acquisition rate. Its effect further increased in wards receiving greater number of patients. In addition, critical care ward, presence of MRSA cohorting beds, ward specialty, and median length of stay were associated with MRSA acquisition.

## Introduction

Since its emergence in the 1960s, Meticillin-Resistant *Staphylococcus aureus* (MRSA) has become endemic in hospitals worldwide, accounting for at least 20% of *Staphylococcus aureus* bloodstream infections globally [1], causing significant health and financial burden [2,3]. In high-income settings, the incidence of hospital-onset MRSA infection has declined over time, although progress in controlling MRSA has plateaued in recent years [4–6].

In Singapore, a high-income city state in Asia, acute care public hospitals initiated a multi-pronged MRSA control strategy from 2006 [7–9] resulting in a substantial reduction in hospital-acquired MRSA bacteremias [7]. Despite the extensive control efforts, MRSA remains endemic in healthcare settings. A point prevalence survey in 2014 indicated that 11.8% of patients in a large tertiary public hospital were colonized by MRSA. The prevalence was higher in intermediate (29.9%), and long-term (20.4%) care facilities [10].

Many factors were found to be associated with MRSA acquisition. They include exposure to other patients known to be colonized with MRSA [11], antibiotic use [11,12], prolonged hospital stay [13–15], and receiving medical procedures during hospitalization [16], intensive care unit (ICU) admission [17], being a trauma or burn injury patient [13,16,17], and alcohol abuse [11]. Other factors include colonization pressure [14,18,19], environmental contamination [20], MRSA colonization status of healthcare staff [21], and organizational factors (such as staff to patient ratio [22,23], bed occupancy rate [24], patient capacity of a ward) [25].

Variation in infection control practices and organizational factors by ward means that intrahospital ward transfer likely change MRSA acquisition risk experienced by a patient. Studies investigating MRSA acquisition risk associated with intra-hospital patient transfer are rare and evidence to date is inconclusive. A case-control study by Dziekan and colleagues found a linear relationship: the greater the number of between-ward transfers, the higher the risk of MRSA acquisition [12]. On the other hand, a prospective cohort study in an acute-care hospital in Brazil, where hospital-wide MRSA surveillance was implemented, showed little evidence of association between ward transfer and MRSA acquisition [26]. This led us to assess whether greater ward connectivity in terms of patient transfer influences the risk of MRSA acquisition. In this study, we used high-resolution electronic medical records of in-patient ward transfers from a large public acute care hospital in Singapore, together with active MRSA admission screening data, to identify ward characteristics associated with MRSA acquisition.

## Methods

We analyzed in-patient electronic medical records from the National University Hospital (NUH), Singapore spanning January 1, 2010 to December 31, 2013. NUH is an acute-care public hospital with more than 1,000 beds. Since 2006, the hospital had implemented a bundle of several MRSA control measures, including (A) active surveillance cultures, (B) hand hygiene promotion; (C) hand hygiene compliance auditing and providing feedback to the wards publicly and to the hospital administration; (D) isolation in a single room or, more commonly,

cohorting MRSA cases in designated cubicles in the wards; (E) other measures: mandating bare below the elbow to all clinical staff, provision of color-coded bracelets for MRSA cases, and contact precaution [7]. These measures are implemented universally in both critical-care (intensive care units or high dependency units) wards and non-critical care wards. Active MRSA screening is not implemented in wards that are considered low risk. These wards include obstetric, pediatric, psychiatric, and acute stay wards.

## Data sources

Data were obtained from three sources.

**Patient affordability simulation system.** Patient Affordability Simulation System (PASS) is a data mart within Singapore regional health system database [27]. The primary function of the regional health system database is to facilitate the population health management initiatives in Singapore. PASS captures hospital service use and cost information of Singapore citizens, permanent residents, and foreigners who sought care at NUH. The following variables were available in the dataset provided to us: ward number, ward specialty, patients' age, and timestamps for patients' admission, ward transfers, and discharge.

**MRSA active surveillance cultures.** Active MRSA screening is implemented in 36 out of 64 in-patient wards. The screening process involves obtaining nasal, axillary, and groin (NAG) swabs at admission, transfer, and discharge. These samples are cultured on selective chromogenic agar. Swabs are obtained on the day of or one day before/after the admission or transfer, and on the day of or one day before discharge. The exceptions are patients hospitalized for shorter than 48 hours, those with a MRSA positive result in a previous hospitalization, and deceased patients. MRSA results from clinical isolates are not captured in the active screening database.

A third-party analyst who was not a study team member linked PASS and MRSA screening datasets using unique patient identifiers and anonymized them before providing access to us. We further linked the screening results to specific instances of admission, transfer, and discharge using swab collection date and time. Missing results between two successive negative results were considered negative.

**Hand-hygiene compliance.** Infection control liaison nurses perform monthly audits in 40 in-patient wards. The audit process includes clandestinely recording twenty observations of healthcare staff hand hygiene activities at random timing [28]. Hand hygiene compliance is defined as per WHO guidance: the number of hand hygiene activities performed as a percentage of the total number of hand hygiene opportunities [29]. Hand-hygiene compliance data were available quarterly for each ward and are linked to PASS at ward level.

## Network analysis

We constructed a weighted directed network using patient transfer data to understand how hospital wards are connected. The network comprised all 64 in-patient wards represented as nodes. Ward connectivity through patient transfers was represented as directed edges linking the origin and destination wards. Edges were also given weights corresponding to the number of patients transferred over a specific period.

To investigate the hypothesis that greater ward connectivity was associated with MRSA acquisition rates, we used in-degree and weighted in-degree as network centrality measures. The former represents the number of other wards from which a focal ward receives at least one patient, while the latter reflects the number of patients a focal ward receives from other wards. We constructed 16 quarterly networks and computed quarterly network measures so that it is consistent with the temporal resolution of hand hygiene compliance data.

## Inclusion and exclusion criteria

We included in the analysis in-patient admissions to one of the 36 active screening wards with hospital stay longer than 48 hours. We defined a hospitalization episode as the period between admission to and discharge from the hospital. One hospitalization episode could contain one or more spells, defined as the period from entry to exit from a hospital ward.

We excluded episodes with a positive or no screening result at admission; episodes of patients younger than 15 (pediatric patients are not routinely screened for MRSA); and episodes with a negative MRSA result at admission but no subsequent MRSA screening results.

A MRSA acquisition event was defined as an initially MRSA-negative patient who was found positive during a hospitalization episode. For each ward, we computed patient-weeks at risk by summing the total time spent by patients in a ward and MRSA acquisition rate (number of acquisitions per 100 patient-weeks). For patients who acquired MRSA, their contribution to patient-weeks at risk was censored at the time of the first positive sample collection.

## Statistical analysis

We used mixed-effects Poisson regression to identify ward-level factors associated with MRSA acquisition. The outcome was the total number of MRSA acquisitions. The natural logarithm of the total patient-weeks at risk was used as an offset. We modelled wards as a random intercept and time (in quarters) as a random slope to account for variability in MRSA acquisition rates by ward and trends, respectively.

We extracted nine explanatory variables. Time-varying variables included ward in-degree and weighted in-degree, number of patients in a ward on a typical day, ward MRSA prevalence among patients directly admitted to the wards and among patients transferred from other wards, length of stay, and hand hygiene compliance. These variables were rescaled before adding to the model; as a result, the unit of each variable corresponded to their standard deviation. Time-invariant variables were critical care ward (i.e., ICU and high dependency unit (HDU)), ward specialty (medical, surgical, orthopedics, oncology, and other), and presence of cohorting beds for MRSA-positive patients.

The number of patients in a ward on a typical day was the quarterly average number of patients registered in each ward on the 15th of each month. This was considered a proxy for a ward's patient capacity.

The colonization pressure of a pathogen in a hospital ward is mainly influenced by the admission of patients who are already colonized [30]. We considered MRSA prevalence among patients directly admitted to the ward and patients transferred from other wards. This allows us to compare their effects on the ward MRSA acquisition rate. In addition, we also assessed the interaction between MRSA prevalence among patients transferred from other wards and weighted in-degree.

**Sensitivity analyses.** As per hospital MRSA screening protocol, if a patient is known to be MRSA-positive, they are not screened in subsequent hospitalizations; consequently, most of those MRSA screening results were absent in the dataset. Therefore, counting the screening results as is would underestimate MRSA prevalence among patients received by a ward. To study its impact on the study results, we imputed the missing MRSA results as positive and compared the study results before and after the imputation.

Hand hygiene compliance audit was not implemented in some of the wards with active MRSA screening and thus data were unavailable. Consequently, we could not include these wards in modelling hand hygiene compliance data. We compared the results of the multivariable models with and without this variable.

In 315 out of 2,608 (12%) episodes with MRSA acquisition events, the screening results were missing in at least one ward spell prior to the spell in which patients were found to be positive. For these episodes, we could not determine the exact ward in which patients acquired MRSA. To assess the impact these episodes on the results, we conducted sensitivity analyses using five scenarios: (1) complete case analysis–we only included episodes with complete screening results for all spells; (2) mid-point analysis–we assumed that MRSA acquisition occurred in the ward the patient was in at the mid-point between the last known negative and the positive result; in the next three scenarios, we probabilistically attributed the acquisition to spells with missing MRSA results by random selection (3) using equal probabilities; (4) using a probability weighted by the patient's length of stay in each spell [13–15,31,32]; and (5) using a probability weighted by both length of stay and overall MRSA prevalence [13,14,31,32]. For scenarios one and two, we obtained point estimates and confidence intervals (CI) from the multivariable model. For scenarios three to five, we iterated the imputation and model fitting 10,000 times to obtain an empirical distribution of the point estimates and 95% CI for each parameter and took the median value.

The analysis based on scenario five after imputing a positive result among hospitalisations of a known MRSA-positive patient, was considered the main analysis as we deemed its assumptions to be more realistically capture the uncertainty associated with missing screening data. Analyses were carried out using R (version 3.5.3) [33]. Network analysis was performed using the igraph package [34] and mixed-effects models were fitted using the lme4 package [35].

### Ethics review

Ethical exemption for this secondary data analysis was obtained from the National Healthcare Group Domain Specific Review Board (reference number: 2018/00890).

## Results

We successfully linked 97.6% of MRSA screening; 2.4% were unlinked because their anonymized identifiers were not found in PASS. A total of 65,428 hospitalization episodes were eligible to investigate factors associated with MRSA acquisition (Fig 1).

### Characteristics of in-patient wards

Of 36 active screening wards, 8 (22%) were critical care wards; 8 (22%) contained MRSA-cohorting beds. Average MRSA prevalence was higher among patients transferred from other wards (10%) compared to directly admitted patients (6.8%). Overall average hand hygiene compliance was 70%, increasing from 64% in 2010 to 73% in 2013. Average in-degree and weighted in-degree were 21 and 151, respectively (Table 1). In-degree was the highest in Ward 1 Surgery (HDU), Ward 1 Medical (ICU/HDU), and Ward 1 Isolation in most quarters over four years. Weighted in-degree was the highest in Ward 1 Surgery (HDU), Ward 3 Cardiac, and Ward 4 Surgery. On the other hand, we observed lowest values in both in-degree and weighted in-degree in Ward 2 Psychiatry, Ward 2 Other, and Ward 5 Coronary care/Cardiac medical.

### MRSA acquisition rates

MRSA acquisitions were identified in 2,608 of 65,428 (4%) hospitalization episodes (Fig 3). In the main analysis, the median overall acquisition rate was 3.5 acquisitions per 100 patient-weeks [95% CI: 3.4, 3.7]. The impact of missing screening results prior to a positive spell on

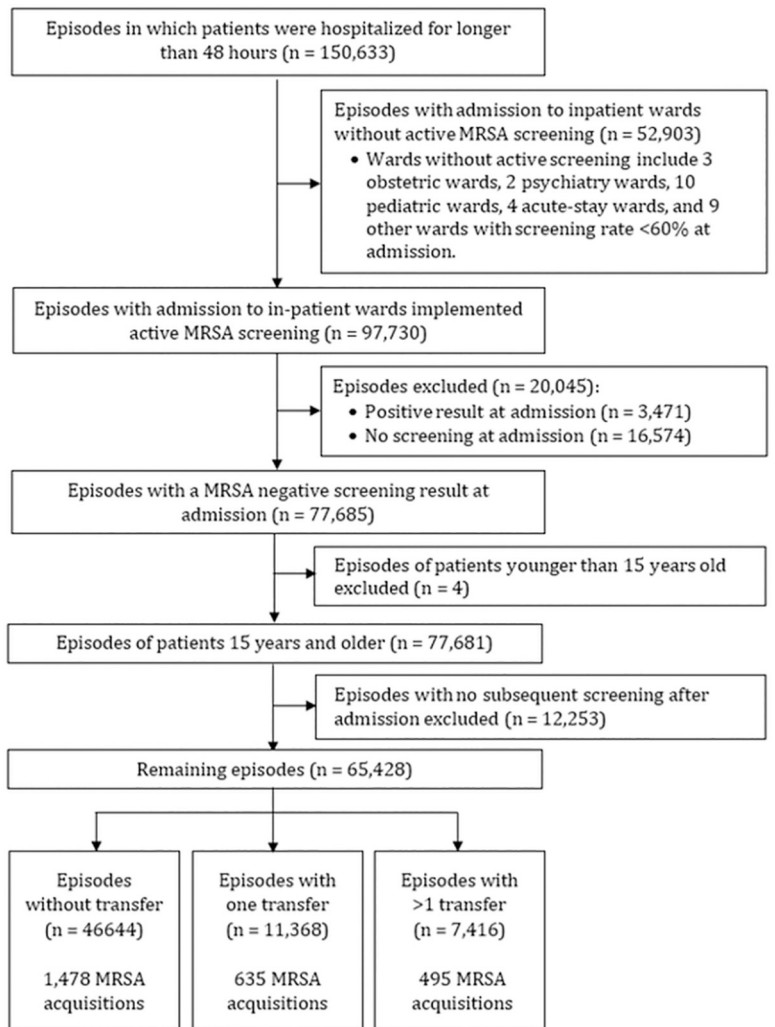

**Fig 1. Hospitalization episodes in National University Hospital, Singapore included in the analysis.**

the estimated acquisition rate was small: the maximum range of variability in 16 quarters over 10,000 iterations was only 0.2 acquisitions per 100 patient-weeks (S1 Fig). The acquisition rates were highest in the hospital wards of the following specialties: surgery, geriatric medicine, orthopedics, and cardiac. Overall MRSA acquisition rates by ward are shown in S1 Table.

## Factors associated with ward-level MRSA acquisition rates

In our main analysis, factors associated with a higher MRSA acquisition rate were: MRSA prevalence among patients transferred from other wards (rate ratio (RR): 7.74 [95% CI: 3.88, 15.44], additional five percentage point increase), critical care ward (RR: 1.72 [95% CI: 1.09, 2.70]) and presence of MRSA cohorting beds in the ward (RR: 1.39 [95% CI: 1.03, 1.90]). On the other hand, oncology ward (RR: 0.66 [95% CI: 0.46, 0.94]) (compared to medical ward), and median length of stay (RR: 0.70 [95% CI: 0.55, 0.90], 1.5 additional days) were associated with a lower acquisition rate (Table 2). In addition, we found evidence of interaction between MRSA prevalence among patients transferred from other wards and weighted in-degree. Fig 2 shows higher number of predicted MRSA acquisitions as the MRSA prevalence among

**Table 1. Characteristics of wards with MRSA active screening at the National University Hospital, Singapore in 2010–2013.**

| Time varying variable | Mean (Standard deviation) * | | | |
|---|---|---|---|---|
| | **2010** | **2011** | **2012** | **2013** |
| Number of patients in a ward on a typical day | 20.1 (±17.2) | 21.7 (±18.6) | 22.5 (±17.7) | 23.5 (±17.8) |
| Length of stay (days) | 3.7 (±1.3) | 4.0 (±1.5) | 3.8 (±1.2) | 4.1 (±1.6) |
| Hand hygiene compliance (%) | 63.9 (±7.1) | 67.1 (±6.8) | 71.1 (±6.4) | 73.3 (±6.2) |
| MRSA prevalence (%) | | | | |
| Among patients directly admitted to the wards | 6.5 (±3.6) | 7.1 (±4.8) | 5.9 (±3.8) | 7.6 (±5.5) |
| Among patients transferred from other wards | 10.5 (±6.5) | 11.2 (±9.7) | 9.3 (±7.9) | 9.1 (±6.3) |
| Measures of ward connectivity | | | | |
| Indegree | 19.4 (±5.9) | 20.3 (±5.3) | 20.9 (±6.6) | 21.9 (±6.2) |
| Weighted indegree | 154.5 (±103.7) | 162.9 (±99.5) | 142.5 (±102.3) | 145.1 (±99.8) |
| **Time invariant variable** | **No. wards** | **%** | | |
| Critical care wards | | | | |
| No | 28 | 88 | | |
| Yes | 8 | 22 | | |
| Presence of MRSA cohorting beds | | | | |
| No | 28 | 88 | | |
| Yes | 8 | 22 | | |
| Ward specialty | | | | |
| Medical | 7 | 19 | | |
| Surgical | 10 | 28 | | |
| Oncology | 8 | 22 | | |
| Orthopedics | 3 | 8 | | |
| Other | 8 | 22 | | |

MRSA, Methicillin-resistant *Staphylococcus aureus*.

* Average values of four quarters were first obtained for each ward. Subsequently, mean and standard deviation of these values are presented.

patients transferred from other wards increases. The rate of increment is higher in wards with greater weighted in-degree. Sensitivity analyses showed that the direction of association was largely consistent across all scenarios (Fig 3).

Imputing a positive result in subsequent hospitalisations of patients with a known MRSA-positive status did not meaningfully changed the results of multivariable analysis except the variables related to MRSA prevalence. Compared to the main model, the model before the imputation showed a higher rate ratio estimate for MRSA prevalence among directly admitted patients and a lower estimate among patients transferred from other wards. Both models showed a higher rate ratio in the latter (Compare results in Tables 2 and S2). In addition, we compared the main model with the one including hand hygiene compliance. In the latter, both unadjusted and adjusted rate ratios showed that hand hygiene compliance itself was not associated with MRSA acquisition rate in the subset of wards in which this information was available, after controlling for other factors. However, compared to the main analysis, the estimates were different for ward specialty, median length of stay, and weighted in-degree (S3 Table).

## Discussion

We used electronic medical records with high temporal resolution to understand in-patient ward connectivity in a large acute care hospital and ward characteristics associated with MRSA acquisition. We found that ward specialty, median length of stay, MRSA prevalence

**Table 2. Ward characteristics associated with MRSA acquisition based on the main analysis.**

| Ward characteristics | Unadjusted RR (95% CI) | Adjusted RR (95% CI) |
|---|---|---|
| Critical care ward | | |
| No | 1 | 1 |
| Yes | 1.06 (0.64, 1.74) | 1.72 (1.09, 2.70) |
| Presence of MRSA cohorting beds | | |
| No | 1 | 1 |
| Yes | 1.53 (0.99, 2.35) | 1.39 (1.03, 1.90) |
| Ward specialty | | |
| Medical | 1 | 1 |
| Oncology | 0.37 (0.23, 0.61) | 0.66 (0.46, 0.94) |
| Ortho | 1.18 (0.63, 2.20) | 0.81 (0.52, 1.29) |
| Other | 1.02 (0.67, 1.55) | 1.21 (0.84, 1.76) |
| Surgery | 0.99 (0.65, 1.51) | 0.91 (0.67, 1.23) |
| MRSA prevalence among directly admitted patients (additional 5 percentage point) | 1.24 (0.89, 1.73) | 0.75 (0.52, 1.09) |
| MRSA prevalence among patients transferred from other wards (one additional 8 percentage point) | 3.99 (2.34, 6.79) | 7.74 (3.88, 15.44) |
| Number of patients on a typical day^ (18 additional patients) | 1.40 (1.11, 1.76) | 1.18 (0.94, 1.50) |
| Median length of stay (1.5 additional days) | 0.75 (0.59, 0.94) | 0.70 (0.55, 0.90) |
| Indegree (one additional ward) | 1.81 (1.01, 3.23) | 1.22 (0.69, 2.18) |
| Weighted-indegree (101 additional patients) | 5.36 (1.45, 19.84) | 2.65 (0.73, 9.68) |
| Interaction term* | | 1.11 (1.01, 1.21) |

CI, Confidence Interval; RR, Rate Ratio.

^ Proxy for ward patient capacity.

* Interaction of MRSA prevalence among transfer patients and weighted in-degree.

among patients transferred from other wards, critical care ward, and presence of MRSA cohorting beds in the ward were associated with MRSA acquisition. However, there is no evidence that ward connectivity measures that we investigated (i.e., indegree, and weighted in-degree) were associated with MRSA acquisition although we observed evidence of interaction between MRSA prevalence among patients transferred from other wards and weighted in-degree.

A ward would have higher MRSA acquisition rate if a high proportion of patients received by the ward are colonised by MRSA. In our results, the evidence is strong that higher MRSA prevalence among transfer patients received by a ward is associated with higher MRSA acquisition rate while MRSA prevalence among directly admitted patients was not associated with MRSA acquisition. This suggests that, on average, MRSA prevalence among patients transferred from other wards had a stronger effect on MRSA acquisition rate, compared to the prevalence among patients directly admitted to the ward. This effect is further increased in wards that received greater volume of patients, as suggested by the interaction between MRSA prevalence among transfer patients and weighted in-degree. In the sensitivity analysis, the results in S2 Table shows that models without accounting for missing MRSA results of patients with a known MRSA-positive status would have overestimated the effect of MRSA prevalence among directly admitted patients while underestimating the effect of MRSA prevalence among patients transferred from other wards. It is worth noting that in this ward-level analysis, weighted in-degree only accounted for the total number of transfers between a ward pair, rather than the total number of transfers experienced by individual patients [12]. For instance, a highly connected ward may have lower acquisition rate, perhaps because of better infection

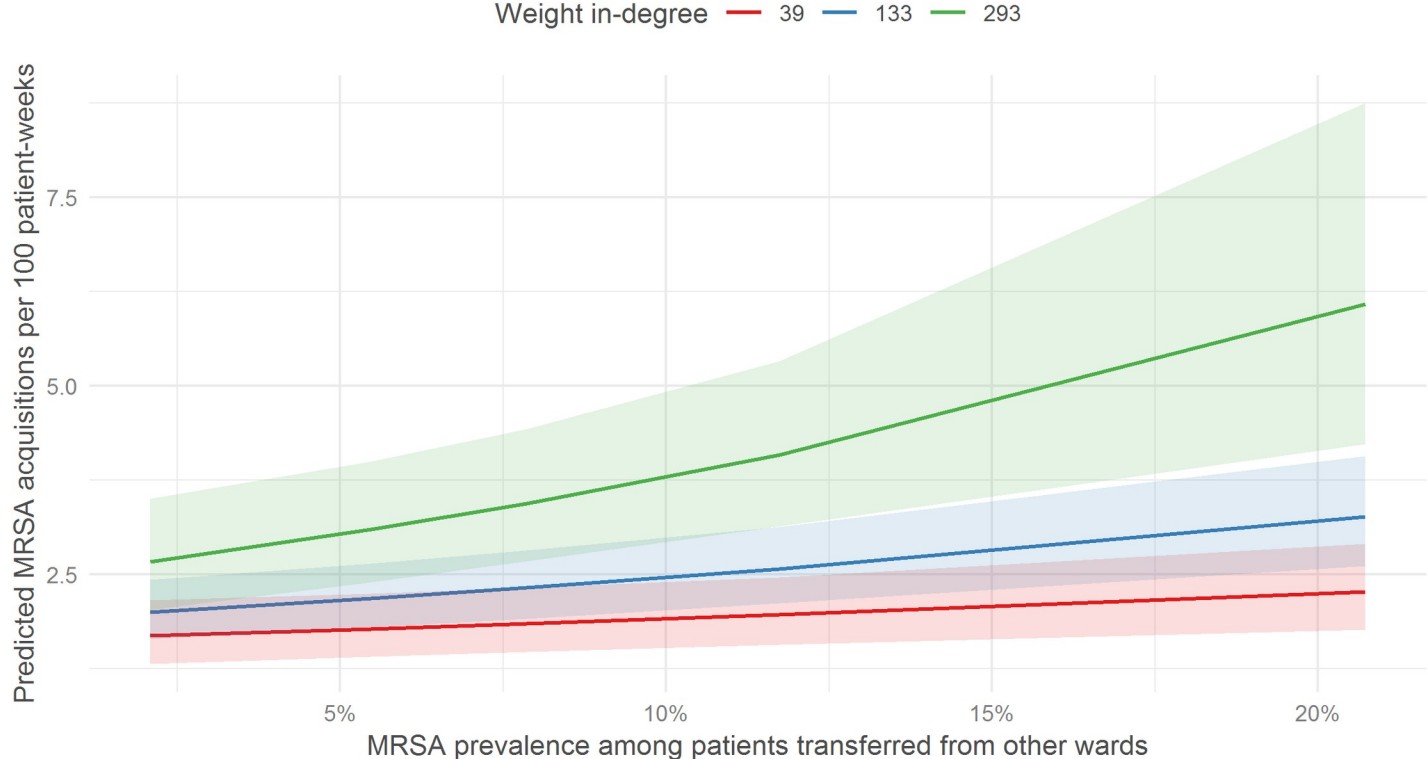

**Fig 2. Predicted MRSA acquisitions and MRSA prevalence among patients transferred from other wards.**

control measures, but it is possible that individual patients from this ward undergoing greater number of transfers may still experience higher MRSA acquisition risk. In this analysis, we investigated two ward connectivity measures deemed to be linked MRSA transmission. However, other network measures may also be relevant.

Although length of hospital stay is an important patient-level risk factor [13–15,31,32], our ward-level analysis showed the opposite: median length of stay was associated with a lower MRSA acquisition rate. A likely explanation is that patients with conditions who required longer hospitalisation tend to be from oncology wards, ICUs and HDUs, the wards in which infection control measures tend to be more stringent. Therefore, MRSA acquisition rates were lower in these wards. Unfortunately, data on infection control measures were unavailable, except hand hygiene compliance data.

MRSA acquisition rate was generally lower in oncology wards compared to other wards, after adjusting for potential confounders. As noted above, the acquisition rate of a ward reflects a balance between the ward's case mix [36] and how stringently infection control measures are implemented [37]. Oncology wards, which tend to have patients at higher risk of infections, are likely to have stricter adherence to infection control measures, and our findings also suggest that improvements in infection control should be possible for other ward types.

Average hand hygiene compliance in NUH was 70% that is comparable to large tertiary hospitals in Hong Kong [38] and Taiwan [39] using similar monitoring protocols. After adjusting for hand hygiene compliance, length of stay was not associated with MRSA acquisition. Hand hygiene compliance data was unavailable in 9 out of 36 eligible wards during the study period. Among them, four were oncology wards and three were ICU/HDU. Patients in these wards tend to require longer hospital stay. The exclusion of these wards from the

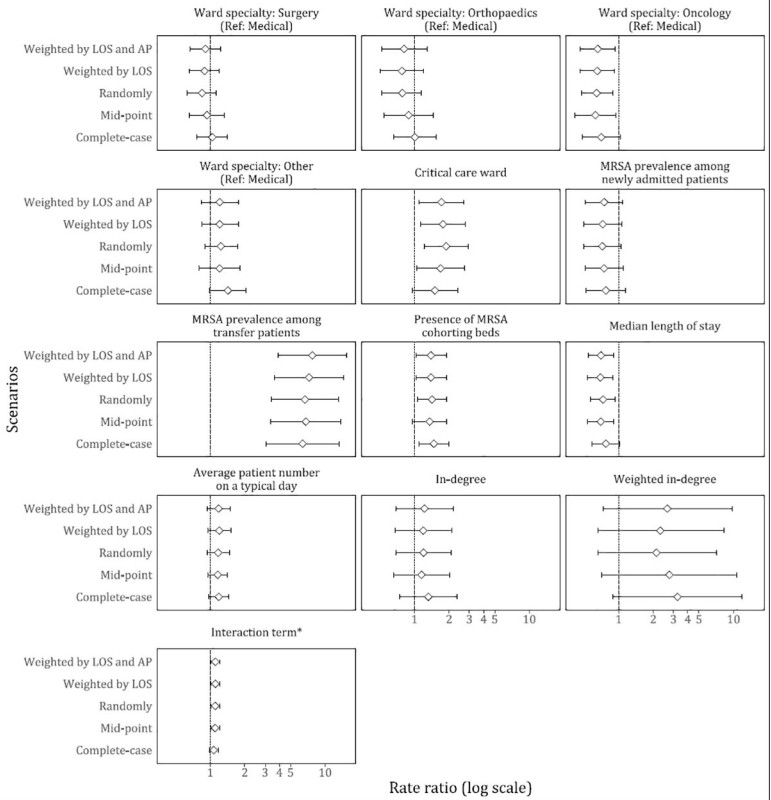

**Fig 3. Sensitivity analyses accounting for the impact of spells without screening prior to a positive spell.** *
Interaction of MRSA prevalence among transfer patients and weighted in-degree. In our main analysis, spells with
missing screening results prior to MRSA acquisition were assigned a positive result with a probability weighted by LOS
and AP of these spells. Each panel describes rate ratio with corresponding 95% confidence interval of each term in the
multivariable models. AP, Admission prevalence; LOS, Length of stay.

multivariable model could explain the lack of association between length of stay and MRSA
acquisition. In addition, effect estimates of oncology also significantly changed in the model
perhaps because almost of half of the excluded wards were of oncology specialty.

Although hand hygiene among healthcare workers is considered the primary infection con-
trol measure in hospital settings, our results showed modest effects of hand hygiene compli-
ance on MRSA acquisition. This could be due to the coarse temporal resolution of quarterly
data that may not accurately capture hand hygiene compliance in the wards. In addition, Haw-
thorne effect may play a role: HCWs may alter their behaviour during the audit, overestimat-
ing hand hygiene compliance. This problem has been previously recognized in NUH [7].

In line with previous studies [36,40,41], we found that critical care ward status was associ-
ated with higher rate of MRSA acquisition. Critical care patients are known to be at particu-
larly high risk for nosocomial infections, pointing to a need for more stringent infection
control measures in these wards. Similarly, the finding that presence of cohorting beds in a
ward is associated with higher MRSA acquisition rate suggests that, despite existing infection
control measures, patients allocated to these beds likely contribute to overall colonisation
pressure.

Several limitations should be considered when interpreting our findings. Firstly, the
unavailability of MRSA results from clinical isolates means that we could not include a subset
of MRSA acquisitions that are not identified through routine screening. However, the

incidence of MRSA infections in NUH is fewer than 1 case per 100 patient-weeks [42], so the impact of this is likely small. Secondly, we could not adjust for ward staffing level [23,43], or MRSA colonization status and compliance with contact precaution measures of healthcare staff as this information is not routinely available [21,44]. Lastly, this ward-level analysis cannot account for individual-level differences in MRSA acquisition risk, including age, gender, comorbidities, and use of out-patient services. More detailed individual-level analyses could investigate the interaction between individual and ward-level risk factors.

Nonetheless, the use of electronic medical records with detailed temporal information on patient transfers and MRSA acquisition within the hospital is a major strength of this analysis. Electronic medical records provide objective measures of patients' transfers that do not rely on recall and self-report.

## Conclusion

Our analysis demonstrates an efficient use of linked electronic medical records and infection control data to comprehensively study the complexity of intrahospital patient transfer patterns. Our findings of ward characteristics associated with MRSA acquisition point to a need for a more targeted approach to improve the current control strategy. In particular, surveillance and control measures should be strengthened in wards with high proportion of MRSA-colonised patients among those transferred from other wards, especially in wards receiving greater volume of transfer patients. Similar methods could be used to understand the transmission dynamics of other nosocomial organisms.

## Supporting information

**S1 Fig. Quarterly MRSA acquisition rate among MRSA active screening wards at National University Hospital, 2010–2013.**
(TIF)

**S1 Table. MRSA acquisition rate in MRSA active screening wards of National University Hospital, 2010–2013.**
(DOCX)

**S2 Table. Results of main analysis before imputing screening results for subsequent hospitalisations of a known MRSA case.**
(DOCX)

**S3 Table. Comparing model results with and without hand hygiene compliance.**
(DOCX)

**S1 File.**
(ZIP)

## Acknowledgments

The authors would like to thank Mark Salloway and Joanne Chee on their assistance in data retrieval. We also thank the healthcare professionals and patients who contributed to the data.

## Author Contributions

**Conceptualization:** Zaw Myo Tun, Clarence C. Tam.

**Data curation:** Sharon Salmon.

**Formal analysis:** Zaw Myo Tun.

**Methodology:** Clarence C. Tam.

**Project administration:** Zaw Myo Tun.

**Resources:** Dale A. Fisher.

**Supervision:** Clarence C. Tam.

**Visualization:** Zaw Myo Tun.

**Writing – original draft:** Zaw Myo Tun.

**Writing – review & editing:** Dale A. Fisher, Sharon Salmon, Clarence C. Tam.

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
