## [Decision Letter · Decision Letter 0]

16 Apr 2021

PONE-D-21-05546

Ward-Level Factors Associated with Methicillin-Resistant Staphylococcus aureus Acquisition – an Electronic Medical Records study in Singapore

PLOS ONE

Dear Dr. Tun,

Thank you for submitting your manuscript to PLOS ONE. After careful consideration, we feel that it has merit but does not fully meet PLOS ONE’s publication criteria as it currently stands. Therefore, we invite you to submit a revised version of the manuscript that addresses the points raised during the review process.

In addition to comments from the reviewers, I have the following comments and suggestions:

1. Describe the baseline infection prevention practices in place for MRSA colonized patients and whether these are the same or different in ICUs vs wards or by specialty.

2. For the finding that longer length of stay is associated with lower risk of MRSA acquisition, the authors speculate that this may reflect stricter infection control measures in the in-patient wards in which patient length of stay is longer. Are there any data to support this? Are longer-stay patients typically housed in certain specific wards? Could this represent bias or confounding in the data? See also the next comment relative to this finding.

3. The authors only brief touch on the significance of the following results: “The results of the latter showed that hand hygiene compliance itself was not associated with MRSA acquisition rate in the subset of wards for which this information was available, after controlling for other factors. However, compared to the main analysis, the estimates were substantially different for ward specialty, presence of MRSA cohorting beds, and median length of stay (S3 Table)”. This merits more discussion, particularly as the length of stay association changed quite a bit, as well as the ward-specific risk estimates after adjusting for hand hygiene compliance. 

4. Supplemental figure 1 shows decrease in acquisition over time. Is there a need to stratify the analysis by time period or otherwise account for temporal changes? Please also see Reviewer 2's comment regarding accounting for different time periods in the analysis.

5. In Table 1, admission prevalence should be expressed as a percentage.

We look forward to receiving your revised manuscript.

Kind regards,

Surbhi Leekha

Academic Editor

PLOS ONE

Journal Requirements:

2. Thank you for stating in the text of your manuscript "A third-party analyst who was not a study team member linked these data sets using unique patient identifiers and anonymized them before providing access to the study team.". Please also add this information to your ethics statement in the online submission form.

Additional Editor Comments (if provided):

Reviewers' comments:

Reviewer's Responses to Questions

**Comments to the Author**

1. Is the manuscript technically sound, and do the data support the conclusions?

Reviewer #1: Yes

Reviewer #2: Yes

2. Has the statistical analysis been performed appropriately and rigorously? 

Reviewer #1: Yes

Reviewer #2: Yes

3. Have the authors made all data underlying the findings in their manuscript fully available?

Reviewer #1: No

Reviewer #2: Yes

4. Is the manuscript presented in an intelligible fashion and written in standard English?

Reviewer #1: Yes

Reviewer #2: Yes

5. Review Comments to the Author

Reviewer #1: This article examines ward-level factors including network measures based on patient transfer patterns to identify ward characteristics that are associated with MRSA acquisition rates. This article adds some expected and unexpected findings to the literature and represents a beginning to further exploration into the ways that patient transfer may or may not contribute to the spread of MRSA across wards within a hospital.

General comment:

It would be surprising if ward connectivity measures as you defined them influenced ward-level MRSA acquisition after controlling for the ward admission prevalence, do you agree? Because what matters is not the ward connectivity at the aggregate level, but how ward connectivity between individual ward-pairs corresponds with the movement of MRSA between wards. But you are not able to capture that in the current way that the model has been structured. Do you have any thoughts about how you may be able to restructure the model so that connectivity measures provide more explicit information about how different wards contribute to acquisition rates via patient transfer?

Specific comments:

Please clarify what ‘the Patient Affordability Simulation System (PASS)” is referring to on lines 13-14 on page 5. Is that the name of the dataset, or the electronic medical record, or a system that interacts with the electronic medical record system?

Please clarify whether or not the “Twenty observations of healthcare staff hand hygiene activities…” (lines 9-10, page 6) are done for each of the 40 random audits each month.

Is there any potential bias in the estimates of hand-hygiene compliance? If so, how might this bias impact your results and conclusions?

Can you add just a brief sentence making more explicit what data were linked? In particular, was the hand-hygiene compliance included in the linked dataset? If so, probably linked at the unit level, but not patient-level, right?

It seems that the Network analysis data is created to correspond with the hand-hygiene compliance data. Are they both included in the ‘Data linkage’ step? If not, perhaps the ‘Data linkage’ paragraph can go after the ‘MRSA Active Surveillance Cultures’ paragraph.

In the calculation of ‘patient-weeks at risk’, the censoring time is the time of collection of a positive MRSA sample. In reality, time at risk could be any time between the last negative test and the first positive test. How might this impact your study?

Clarify how ward MRSA admission prevalence is computed (lines 9-15, page 8). Is it the number of positive tests at ward admission/transfer divided by the total number of tests that occur at ward admission/transfer?

Why were you unsuccessful at linking 100% of MRSA screening results to PASS (lines 18-19, page 9)?

Reviewer #2: The authors study the co-relations/associations between MRSA cases (or acquisition rates) and ward characteristics, including the network weighted in-degrees, given by patient transfers.

This is a valid and relevant study. I have a few technical suggestions/comments:

- Data spans 4 years and were divided into quarters. It is unclear to me how the authors took into account these different periods in the analysis. The ward population varies considerable over the year and this may have an effect in terms of screening, for example. If we look at table 2, the confidence intervals are quite broad suggesting a large variance in numbers. Reporting data for different quarters (e.g. 4 quarters, with averages per quarter in different years) or different years (e.g. averages over all quarters in the same year) might be more appropriate for a comparative analysis and to reduce the effect of high variation of inpatient flow.

- In the discussion, it could be pointed out that other network structures may be relevant as indicators or to be associated with MRSA cases. In-degree is an important but very simple network measure.

-similarly, the abstract reads "We did not find evidence that network measures of 20 ward connectivity, including in-degree, weighted in-degree, ..." But in reality, the authors study only in-degree and weighted in-degree. This sentence (and also "relative connectivity" in the conclusion) should be rewritten to make this point clear.

- Dynamic contact networks of patients and MRSA spread in hospitals. Sci Rep 10, 9336 (2020). https://doi.org/10.1038/s41598-020-66270-9 is very much related to this study and could help improving the contextualization in the introduction and findings reported in the discussion section.

minor:

- I suggest the authors review the text to remove a few typos here and there and improve a little the grammar

- the CI is not always reported in a standard format (see e.g. p12, row 4), I suggest to always report using brackets []

- In p6, "<48 hours" should be "in less than 48 hours". Review other cases along the text.

6. PLOS authors have the option to publish the peer review history of their article (what does this mean?). If published, this will include your full peer review and any attached files.

Reviewer #1: No

Reviewer #2: No

---

## [Author Response · Author response to Decision Letter 0]

14 Jun 2021

Response to reviewers’ comments

We thank the reviewers and the editor for providing useful comments. We have made substantial changes to the analysis based on these comments and beyond.

We updated the measurement of admission prevalence. We previously measured admission prevalence using the first hospitalisation of all patients between 2010 and 2013 as screening is not performed in subsequent hospitalisations of a known positive patients, as per the protocol. In the updated strategy, we assessed MRSA prevalence among patients directly admitted to the ward and among patients transferred from other wards. This approach allowed us to assess the relative contribution of each type of patients received by a ward on MRSA acquisition. In addition, we were able to assess the interaction between MRSA prevalence among patients transferred from other wards and weighted in-degree. This is related to point number 6 below.

As per hospital screening protocol, patients who are known to be MRSA-positive are not screening for MRSA in subsequent hospitalisations; the screening results were absent in most of these hospitalisations. To study its impact on the study results, we compared the model results before and after imputing those absent results as positive.

We also updated the linkage procedure and linked the MRSA screening results that were previously unlinked. The proportion of the screening results successfully linked increased from 92% to 97.6%. Please see point 14 for more details.

Finally, results of continuous variables in the multivariable models were not correctly interpreted previously. Specifically, these variables were rescaled before they were added to the model. Consequently, the unit of these variables corresponded to their standard deviation. This has been correctly reflected in the revision.

Please see our point-to-point responses below.

1. Describe the baseline infection prevention practices in place for MRSA colonized patients and whether these are the same or different in ICUs vs wards or by specialty.

Response: Details have been added as suggested under ‘Methods’ section. (Lines 10-19, page 4)

2. For the finding that longer length of stay is associated with lower risk of MRSA acquisition, the authors speculate that this may reflect stricter infection control measures in the in-patient wards in which patient length of stay is longer. Are there any data to support this? Are longer-stay patients typically housed in certain specific wards? Could this represent bias or confounding in the data? See also the next comment relative to this finding.

Response: Patients who require longer hospitalisation tend to be in oncology wards, intensive care and high dependency units (ICU/HDU). The median length of stay was longer in oncology wards or ICU/HDU (4 days or longer) than other wards, based on S5 Table in the revised manuscript. Infection control measures tend to be more stringent in these wards. This is a likely explanation of the association. Unfortunately, data on infection control measures (e.g., MRSA colonization status and compliance with contact precaution measures of healthcare staff) were unavailable, except the hand hygiene compliance.

3. The authors only brief touch on the significance of the following results: “The results of the latter showed that hand hygiene compliance itself was not associated with MRSA acquisition rate in the subset of wards for which this information was available, after controlling for other factors. However, compared to the main analysis, the estimates were substantially different for ward specialty, presence of MRSA cohorting beds, and median length of stay (S3 Table)”. This merits more discussion, particularly as the length of stay association changed quite a bit, as well as the ward-specific risk estimates after adjusting for hand hygiene compliance.

Response: Hand hygiene compliance data was unavailable in 9 out of 36 eligible wards during the study period. Among them, 4 were oncology wards and 3 were ICU/HDU. As noted in previous point, patients in these wards tend to require longer hospital stay. Excluding these wards from the multivariable model would likely reduce the effect of length of stay on MRSA acquisition. This could explain the change in effect estimate in median length of stay.

In addition, the significant change in the effect estimate for oncology wards could be because almost half of the excluded wards were oncology specialty.

4. Supplemental figure 1 shows decrease in acquisition over time. Is there a need to stratify the analysis by time period or otherwise account for temporal changes? Please also see Reviewer 2's comment regarding accounting for different time periods in the analysis.

Response: We accounted for the temporal change in MRSA acquisition rate by modelling time (quarters) as a random slope in the mixed-effects Poisson models. The random slope model allows the degree of decline in MRSA acquisition over time to vary across in-patient wards, thus adjusting for secular changes in acquisition rate between wards.(1)

5. In Table 1, admission prevalence should be expressed as a percentage.

Response: We have measured MRSA prevalence explicitly in both among newly admitted patients and among transfer patients, as described above. We have expressed both as percentages in table 1.

Reviewer #1: This article examines ward-level factors including network measures based on patient transfer patterns to identify ward characteristics that are associated with MRSA acquisition rates. This article adds some expected and unexpected findings to the literature and represents a beginning to further exploration into the ways that patient transfer may or may not contribute to the spread of MRSA across wards within a hospital.

6. It would be surprising if ward connectivity measures as you defined them influenced ward-level MRSA acquisition after controlling for the ward admission prevalence, do you agree? Because what matters is not the ward connectivity at the aggregate level, but how ward connectivity between individual ward-pairs corresponds with the movement of MRSA between wards. But you are not able to capture that in the current way that the model has been structured. Do you have any thoughts about how you may be able to restructure the model so that connectivity measures provide more explicit information about how different wards contribute to acquisition rates via patient transfer?

Response: Thank you for these insightful comments. We agree that total ward transfers alone cannot adequately capture the MRSA transmission through patient transfers. In our analysis, it is not possible to explicitly model the number of MRSA-colonised patients received by a focal ward from each of the other wards because it would require a larger number of parameters than there are observations.

Instead, we separately measured MRSA prevalence among patients directly admitted into each ward and patients transferred from other wards and assessed their effects on MRSA acquisition rate. This would provide information on the relative contribution of each measure on MRSA acquisition. In addition, we were able to examine the interaction between MRSA prevalence among transfer patients and weighted in-degree.

7. Please clarify what ‘the Patient Affordability Simulation System (PASS)” is referring to on lines 13-14 on page 5. Is that the name of the dataset, or the electronic medical record, or a system that interacts with the electronic medical record system?

Response: We have added details of PASS as suggested. (Lines 23, page 4)

8. Please clarify whether or not the “Twenty observations of healthcare staff hand hygiene activities…” (lines 9-10, page 6) are done for each of the 40 random audits each month.

Response: We have revised the sentence for clarity. Twenty observations of healthcare staff hand hygiene activities are recorded at random timing in each of 40 in-patient wards each month.

9. Is there any potential bias in the estimates of hand-hygiene compliance? If so, how might this bias impact your results and conclusions?

Response: Hawthorne effect may play a role in hand hygiene compliance measurement: healthcare staff may alter their behaviour if they know that they are being audited. This would overestimate hand hygiene compliance. This problem was previously recognized in NUH (2).

This has been added to the manuscript. (Line 2, page 18)

10. Can you add just a brief sentence making more explicit what data were linked? In particular, was the hand-hygiene compliance included in the linked dataset? If so, probably linked at the unit level, but not patient-level, right?

Response: Indeed, hand-hygiene compliance data were available for each quarter for each ward and therefore, they are linked at ward level. We have added this clarification in the manuscript. (Line 23, page 5)

11. It seems that the Network analysis data is created to correspond with the hand-hygiene compliance data. Are they both included in the ‘Data linkage’ step? If not, perhaps the ‘Data linkage’ paragraph can go after the ‘MRSA Active Surveillance Cultures’ paragraph.

Response: Thank you. We have moved the ‘data linkage’ paragraph for MRSA screening and hand-hygiene compliance data accordingly based on your suggestion.

12. In the calculation of ‘patient-weeks at risk’, the censoring time is the time of collection of a positive MRSA sample. In reality, time at risk could be any time between the last negative test and the first positive test. How might this impact your study?

Response: Indeed, MRSA acquisition could have happened any time between the last negative and the first positive results. Our time censoring approach would likely underestimate the acquisition rate. The longer the duration between the sample collection of the last negative and the first positive results, the more variable in the degree of the underestimation.

At the same time, longer hospital stay is also a known patient-level risk factor for MRSA acquisition. This implies that the time of MRSA acquisition may be closer to the end of their ward stay. This is common in wards with patients whose conditions require longer hospital stay (for example, oncology).

13. Clarify how ward MRSA admission prevalence is computed (lines 9-15, page 8). Is it the number of positive tests at ward admission/transfer divided by the total number of tests that occur at ward admission/transfer?

Response: In the revised manuscript, MRSA prevalence was estimated among both directly admitted patients and patients transferred from other wards. This has been clarified in the manuscript. (Lines 21-23, page 7)

14. Why were you unsuccessful at linking 100% of MRSA screening results to PASS (lines 18-19, page 9)?

Response: We revisited data linkage mechanisms and found that we could perform linkage on previously unlinked MRSA screening results. The percentage of successfully linked screening results has increased to 97.6%. However, we still could not link the remaining 2.4% because their anonymized unique identifiers could not be found in PASS.

The previous linkage procedure only considered the MRSA screening results that could be mapped directly to a patient movement date with a buffer period of 24 hours before and after the date. This approach left out the results that did not fall within the buffer period. We have updated the procedure and linked the remaining results.

Reviewer #2: The authors study the co-relations/associations between MRSA cases (or acquisition rates) and ward characteristics, including the network weighted in-degrees, given by patient transfers.

This is a valid and relevant study. I have a few technical suggestions/comments:

15. - Data spans 4 years and were divided into quarters. It is unclear to me how the authors took into account these different periods in the analysis. The ward population varies considerable over the year and this may have an effect in terms of screening, for example. If we look at table 2, the confidence intervals are quite broad suggesting a large variance in numbers. Reporting data for different quarters (e.g. 4 quarters, with averages per quarter in different years) or different years (e.g. averages over all quarters in the same year) might be more appropriate for a comparative analysis and to reduce the effect of high variation of inpatient flow.

Response: As noted in point 4, the effect of time period on MRSA acquisition is accounted for by using time as the random slope parameter in the mixed-effects Poisson model.(1) The random slope parameter adjusts for the variability of acquisition rate by ward over time.

For reporting time varying variables, mean and standard deviation have been reported for each calendar year in table 1 as suggested.

16. In the discussion, it could be pointed out that other network structures may be relevant as indicators or to be associated with MRSA cases. In-degree is an important but very simple network measure.

Response: We have added this point in the discussion. (Line 1, page 17) Thanks for the suggestion.

17. Similarly, the abstract reads "We did not find evidence that network measures of 20 ward connectivity, including in-degree, weighted in-degree, ..." But in reality, the authors study only in-degree and weighted in-degree. This sentence (and also "relative connectivity" in the conclusion) should be rewritten to make this point clear.

Response: We have amended the abstract accordingly.

18. Dynamic contact networks of patients and MRSA spread in hospitals. Sci Rep 10, 9336 (2020). https://doi.org/10.1038/s41598-020-66270-9 is very much related to this study and could help improving the contextualization in the introduction and findings reported in the discussion section.

Response: Thank you for your suggestion. This paper simulated the spread of MRSA within and between a network of healthcare facilities in the Stockholm County in Sweden. In the simulation, the study team incorporated patient contact patterns based on a dataset with rich temporal resolution, a feature similar to the dataset used in our current analysis. Contact was defined as a patient pair sharing a ward at the same time.

However, our focus is to identify network features and other factors at ward level, rather than patient-level, that are associated with MRSA acquisition. Therefore, the contact network described in the paper is not directly related to our study although it is certainly a useful reference for our patient-level network analysis that is underway.

19. Minor:

- I suggest the authors review the text to remove a few typos here and there and improve a little the grammar

- the CI is not always reported in a standard format (see e.g. p12, row 4), I suggest to always report using brackets []

- In p6, "<48 hours" should be "in less than 48 hours". Review other cases along the text.

Response: Thank you. We have made the changes as suggested.

References

1. Luke DA. Multilevel modeling. Thousand Oaks, Calif: Sage Publications; 2004. 79 p. (Sage university papers. Quantitative applications in the social sciences). 

2. Fisher D, Tambyah PA, Lin RT, Jureen R, Cook AR, Lim A, et al. Sustained meticillin-resistant Staphylococcus aureus control in a hyper-endemic tertiary acute care hospital with infrastructure challenges in Singapore. J Hosp Infect. 2013/09/10 ed. 2013 Oct;85(2):141–8.

---

## [Editor Report · Decision Letter 1]

5 Jul 2021

Ward-Level Factors Associated with Methicillin-Resistant Staphylococcus aureus Acquisition – an Electronic Medical Records study in Singapore

PONE-D-21-05546R1

Dear Dr. Tun,

We’re pleased to inform you that your manuscript has been judged scientifically suitable for publication and will be formally accepted for publication once it meets all outstanding technical requirements.

Kind regards,

Surbhi Leekha

Academic Editor

PLOS ONE
---

## [Editor Report · Acceptance letter]

8 Jul 2021

PONE-D-21-05546R1 

Ward-Level Factors Associated with Methicillin-Resistant *Staphylococcus aureus* Acquisition – an Electronic Medical Records study in Singapore 

Dear Dr. Tun:

I'm pleased to inform you that your manuscript has been deemed suitable for publication in PLOS ONE. Congratulations! Your manuscript is now with our production department. 

Kind regards, 

on behalf of

Dr. Surbhi Leekha 

Academic Editor

PLOS ONE